# Ultrasensitive detection of local acoustic vibrations at room temperature by plasmon-enhanced single-molecule fluorescence

Mingcai Xie[1], Hanyu Liu[1], Sushu Wan [1], Xuxing Lu[1,2], Daocheng Hong[1], Yu Du[1], Weiqing Yang[1], Zhihong Wei[1], Susu Fang[1], Chen-Lei Tao[3], Dan Xu[3], Boyang Wang[4], Siyu Lu[4], Xue-Jun Wu [3], Weigao Xu [1], Michel Orrit [2✉] & Yuxi Tian[1✉]

Sensitive detection of local acoustic vibrations at the nanometer scale has promising potential applications involving miniaturized devices in many areas, such as geological exploration, military reconnaissance, and ultrasound imaging. However, sensitive detection of weak acoustic signals with high spatial resolution at room temperature has become a major challenge. Here, we report a nanometer-scale system for acoustic detection with a single molecule as a probe based on minute variations of its distance to the surface of a plasmonic gold nanorod. This system can extract the frequency and amplitude of acoustic vibrations with experimental and theoretical sensitivities of 10 pm Hz$^{-1/2}$ and 10 fm Hz$^{-1/2}$, respectively. This approach provides a strategy for the optical detection of acoustic waves based on molecular spectroscopy without electromagnetic interference. Moreover, such a small nano-acoustic detector with 40-nm size can be employed to monitor acoustic vibrations or read out the quantum states of nanomechanical devices.

[1] Key Laboratory of Mesoscopic Chemistry of MOE, School of Chemistry and Chemical Engineering, Nanjing University, Nanjing 210023, China. [2] Huygens-Kamerlingh Onnes Laboratory, Leiden University, 2300 RA Leiden, The Netherlands. [3] State Key Laboratory of Coordination Chemistry, School of Chemistry and Chemical Engineering, Nanjing University, Nanjing 210023, China. [4] Green Catalysis Center, and College of Chemistry, Zhengzhou University, Zhengzhou 450001, China. ✉email: orrit@physics.leidenuniv.nl; tyx@nju.edu.cn

Mechanical waves, especially acoustic waves, play essential roles in our daily communications and have facilitated many advanced technologies, such as sonar[1], seismography[2], and ultrasound imaging[3], owing to their efficient propagation even in moderately heterogeneous media. Thus, the detection of acoustic waves is fundamental in these techniques and their applications. Considerable effort has been made to improve the detection sensitivity of these technologies. With the rapid development of nanotechnology, diverse technological devices have appeared, such as micro-electromechanical systems (MEMS)[4,5], nano-electromechanical systems (NEMS)[6,7], optical tweezers[8,9], AFM nanocantilevers[10,11], nanomechanical mass spectrometers[12,13], quantum-optomechanical devices[14,15], and surface-acoustic-wave (SAW) resonators[16,17], which are used to detect acoustic vibrations and displacements at the nanometer scale. However, it is difficult to simultaneously achieve high detection sensitivity and high spatial resolution. Therefore, fulfilling high-sensitivity detection of acoustic vibrations with high spatial resolution is of great importance and has potential applications for high-resolution ultrasound imaging, accurate localization of the defects in materials and sensitive tracking of acoustic sources.

Single molecules are widely used as fluorescence probes for positioning and optical imaging at high resolution (1 Å)[18,19]. The spectral properties of a single molecule strongly depend on the surrounding environment in addition to the intrinsic chemical structure of the molecule[20,21]. For example, the zero-phonon line of a single molecule is very sensitive to the electric field, pressure, and stress at low temperature[22]. Based on this large spectral sensitivity, some of us have successfully achieved the detection of weak acoustic vibrations with high sensitivity and resolution by a single molecule[23]. However, this acoustic detection method relying on lifetime-limited optical resonances is restricted to low-temperature conditions, limiting its applications. Instead of the spectral sensitivity, the very steep dependence of fluorescence enhancement on the distance between a molecule and a plasmonic structure is proposed in this study.

It has been widely reported that fluorescence can be significantly enhanced or quenched by the localized surface plasmon resonance of metal nanoparticles[24–26]. The fluorescence enhancement is strongly dependent on the distance between a molecule and a plasmonic nanoparticle[27–29]. Thus, the enhanced fluorescence signal can be adopted to sensitively detect acoustic vibrations modulating the distance between the molecule and the nanoparticle. In this work, the fluorescence signal of a single molecule of the dye crystal violet (CV) located close to a gold nanoparticle is tracked, and the specific frequency and amplitude of acoustic vibrations are successfully extracted at room temperature.

## Results

**Arrangement of plasmon-molecule nano-detector**. The single-molecule acoustic detector is formed by a single CV molecule located close to a gold nanorod as indicated in Fig. 1a. Since the large fluorescence enhancement of molecules with low quantum yield (QY) provides better contrast against the emission background[24,30], crystal violet (CV, see the inset in Fig. 1b for chemical structure) with QY of 0.5% measured in the PMMA film was selected as a probe to detect acoustic vibrations. Good photostability is a basic requirement for molecules as fluorescent probes. Thus, we first checked the photostability of CV molecules at the ensemble-averaged and single-molecule levels. As indicated in Fig. 1b, at the ensemble-averaged level, the CV-doped film shows fluorescence enhancement at the initial stage of 1 min, and keeps unchanged for 10 min. The result indicates excellent

photostability of CV molecules in the film. At the single-molecule level, we recorded histograms of on/off-times as a characterization of single-molecule bleaching, as illustrated in Fig. 1c and Supplementary Note 2. The result shows that, although most CV molecules can be bleached, still many molecules can survive within 10 min. To obtain stable and reliable acoustic signals, we automatically selected the surviving ones for the acoustic detection measurements. The morphologies of the purchased gold nanorods were characterized by SEM, as illustrated in the inset of Fig. 1d. Their average length and diameter were ~35 and 16 nm (aspect ratio of 2.2), respectively, as presented in Fig. 1d and Supplementary Note 3. The absorption spectrum of a suspension of gold nanorods, absorption, and fluorescence spectra of CV embedded in the PMMA film are exhibited in Fig. 1e. Notably, the longitudinal plasmonic resonance peak of gold nanorods is at 640 nm, which largely overlaps with the fluorescence and absorption spectra of CV. This significant overlap guarantees a high fluorescence enhancement factor for the CV fluorescence. The second peak at 520 nm corresponds to a small number of gold spheres in the sample during the fabrication of gold nanorods. It is worth noting that a 633 nm laser was adopted in this work to selectively excite individual gold nanorods on their longitudinal plasmonic resonance (SPR around 633 nm), so as to minimize the influence from the gold spheres. For the acoustic detection measurements, as shown in the schematic of Fig. 1a, the molecule-nanorod systems were embedded in a PMMA thin film as the medium. Then, the PMMA film was fixed to the two prongs of a quartz crystal tuning fork. The choice of a medium with proper rigidity is also imperative. The medium should be easily deformed by the acoustic wave and be able to transfer acoustic waves properly. Additionally, the medium should be transparent to light and easily teared off the substrate. After experimenting with different polymers including PMMA, PS, PDMS, and PVA, we found that PMMA is the most suitable medium for this measurement.

**Characterization and extraction of acoustic signals**. In the measurement, a quartz crystal tuning fork generated an acoustic wave at a fixed frequency on its resonance. The resonant frequency of the tuning fork was characterized (Supplementary Note 4) before each measurement to ensure that the tuning fork was well driven. The quality factor Q of the tuning fork without and with the sample was characterized to be 5000 and 2000, respectively. The compression and extension of the PMMA film were caused in the direction perpendicular to the prongs of the tuning fork when the driving voltage was set at the resonant frequency of the tuning fork. As a result, the distance between the CV molecule and the gold nanorod was periodically changed, leading to periodic variations in the fluorescence intensity. In our experiments, the acoustic wave was detected through bright spots with high fluorescence intensity because only the molecules located close to the gold nanorods can be detected. This can be verified by the coincidence between the fluorescence and scattering images (see Supplementary Note 5). The significantly shortened lifetime also supports the enhancement effect (see Supplementary Note 5). Figure 2a illustrates a typical fluorescence image of a single CV molecule embedded in the film of the tuning fork that is largely enhanced by the gold nanorod. The fluorescence intensity was recorded over 200 s based on time-correlated single-photon counting when the tuning fork was driven at the resonant frequency of 32.710 kHz. Besides, a fast Fourier transform (FFT) of the intensity traces was performed to obtain the information of the intensity variation as shown in Fig. 2b. A sharp peak at the tuning fork's resonant frequency of 32.710 kHz can be clearly observed, and this peak value is linearly proportional to

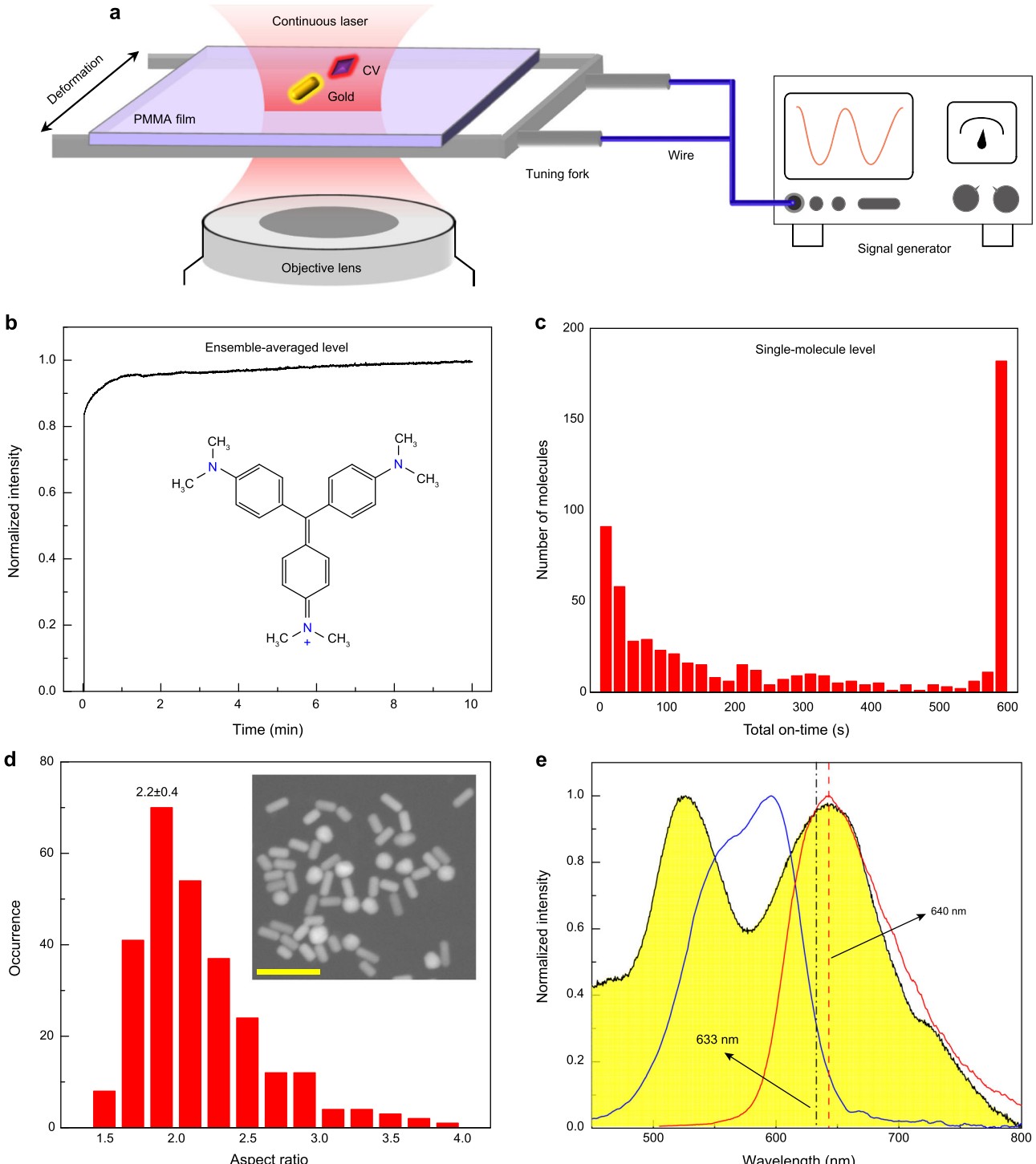

**Fig. 1 Characterization of a single molecule/gold nanorod detection system for acoustic vibrations. a** Schematic of the nano-acoustic detector based on the distance variation between the molecule and the nanorod. The molecule-nanorod detection system was first embedded in the PMMA film, which was attached on the two prongs of the tuning fork. Then, the tuning fork was driven at the resonant frequency by a signal generator and vibrated periodically. **b** The fluorescence intensity traces of crystal violet (CV) molecules film. Inset: the chemical structure of CV. **c** Statistics of total on-state duration time of individual CV molecules. **d** Characteristic distribution of the geometric size of gold nanorods. Their aspect ratio (length/diameter) is 2.2 ± 0.4. Inset: SEM image of a drop of gold nanorod suspension dried on a silicon chip; Scale bar: 100 nm. **e** Absorption spectrum of a suspension of gold nanorods (yellow shaded area), absorption (blue solid line), and fluorescence (red solid line) spectra of CV in the PMMA film. There is a significant spectral overlap between CV molecules and gold nanorods. The vertical black dashed-dotted line is the excitation light source guided by the black solid arrow, and the vertical red dashed line is the peak of the fluorescence spectrum of CV guided by the black solid arrow.

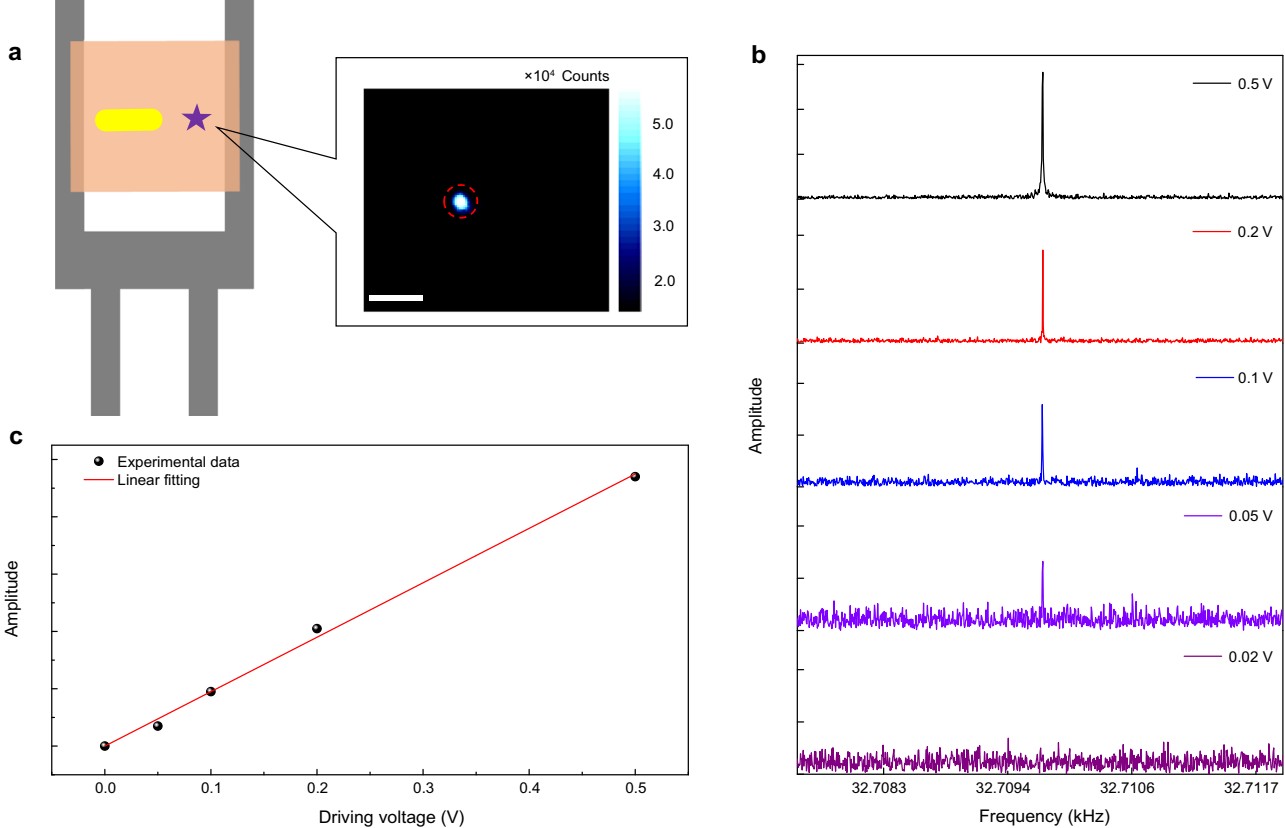

**Fig. 2 Experimental detection of acoustic vibrations. a** Schematic of the arrangement of the experimental system. Inset shows the fluorescence image of a single crystal violet (CV) molecule; scale bar: 5 μm. **b** Fast Fourier transform (FFT) of the fluorescence intensity traces (with a bin size of 5 μs for a 200-s acquisition time) when driven with shifting voltages of 0.02−0.5 V at the resonant frequency. The significant sharp peak at 32.710 kHz indicates perfect modulation of the optical signal by the tuning fork. The limit voltage for a detectable signal is 0.05 V. **c** The FFT amplitude as a function of the driving voltage. The FFT peak is linearly proportional to the driving voltage. The fluorescence intensity was kept constant (~26,000 counts s$^{-1}$) for different driving voltages. The size of a CV molecule is about 2 nm. The red solid line is the linear fitting of the experimental data (solid scattered points).

the driving voltage, as observed in Fig. 2c. Besides, we also provide another two molecules with acoustic response and investigate their FFT peaks as functions of the driving voltage, as presented in Supplementary Note 6. Although a small deviation from a linear fitting occurs in one molecule, as a whole, the FFT peaks in both of molecules vary linearly with the driving voltage.

**Calculation of experimental sensitivity of nano-detector.** The detection sensitivity was characterized by estimating the minimum detectable vibration amplitude. The tuning fork was imaged under a bright-field microscope without and with a driving voltage of 15 V to visualize the deformation of the tuning fork prong, as illustrated in Figs. 3a and b, respectively. The displacement can be directly characterized from the image, as illustrated in Fig. 3c. As revealed by varying the driving voltage, the displacement of the tuning fork prong is linearly related to the driving voltage, with a slope of ~200 nm V$^{-1}$ (Fig. 3d). Thus, the detection limit of the displacement is calculated to be 10 nm corresponding to a driving voltage of 0.05 V. The PMMA film is stretched by 20 nm, which corresponds to a relative deformation factor of $7 \times 10^{-5}$ according to the 300-μm gap between the two prongs. Consequently, the detection sensitivity of a single CV molecule with a distance of approximately 10 nm towards a gold nanorod can be estimated to be 10 pm Hz$^{-1/2}$ under the consideration of an acquisition time of 200 s. Particularly, the detection sensitivity of each molecule is different which depends on the molecule-nanorod distance as well as on the orientation of the system. Less than one-third of the

molecules have high enough detection sensitivity to sense the acoustic wave generated by the tuning fork driven at 5 V. Furthermore, another single molecule with a detection sensitivity of 20 pm Hz$^{-1/2}$ is introduced in Supplementary Note 7, and the statistical distributions of experimental detection sensitivity for 24 molecules are given in Supplementary Note 8. Another crucial parameter for an acoustic detector is bandwidth. The bandwidth of the molecule-nanorod system depends on the time resolution of our photon-counting instrument, which is 4 ps, corresponding to a bandwidth of 40 GHz when considering 6 points per oscillation according to Nyquist's criterion (see Supplementary Note 9). Actually, the bandwidth can be further improved by employing a spectrum analyser which breaks the Nyquist's criterion[31]. It is worth noting that to achieve such broad bandwidth, we need to excite the molecule many times to collect enough photons, which limits our method's time resolution for real-time measurements. The bandwidth of the real-time measurements is limited by the fluorescence rate of the molecule which is about 4 kHz for the current system. Molecules with shorter lifetime or better optimization could provide a broader bandwidth.

**Calculation of theoretical sensitivity of nano-detector.** The detection sensitivity of such a single molecule-nanorod system can be theoretically estimated from the plasmonic enhancement effect based on a finite-element method (FEM). The incident light interacting with metal nanoparticles can induce a collective oscillation of free electrons when the incident light is on the

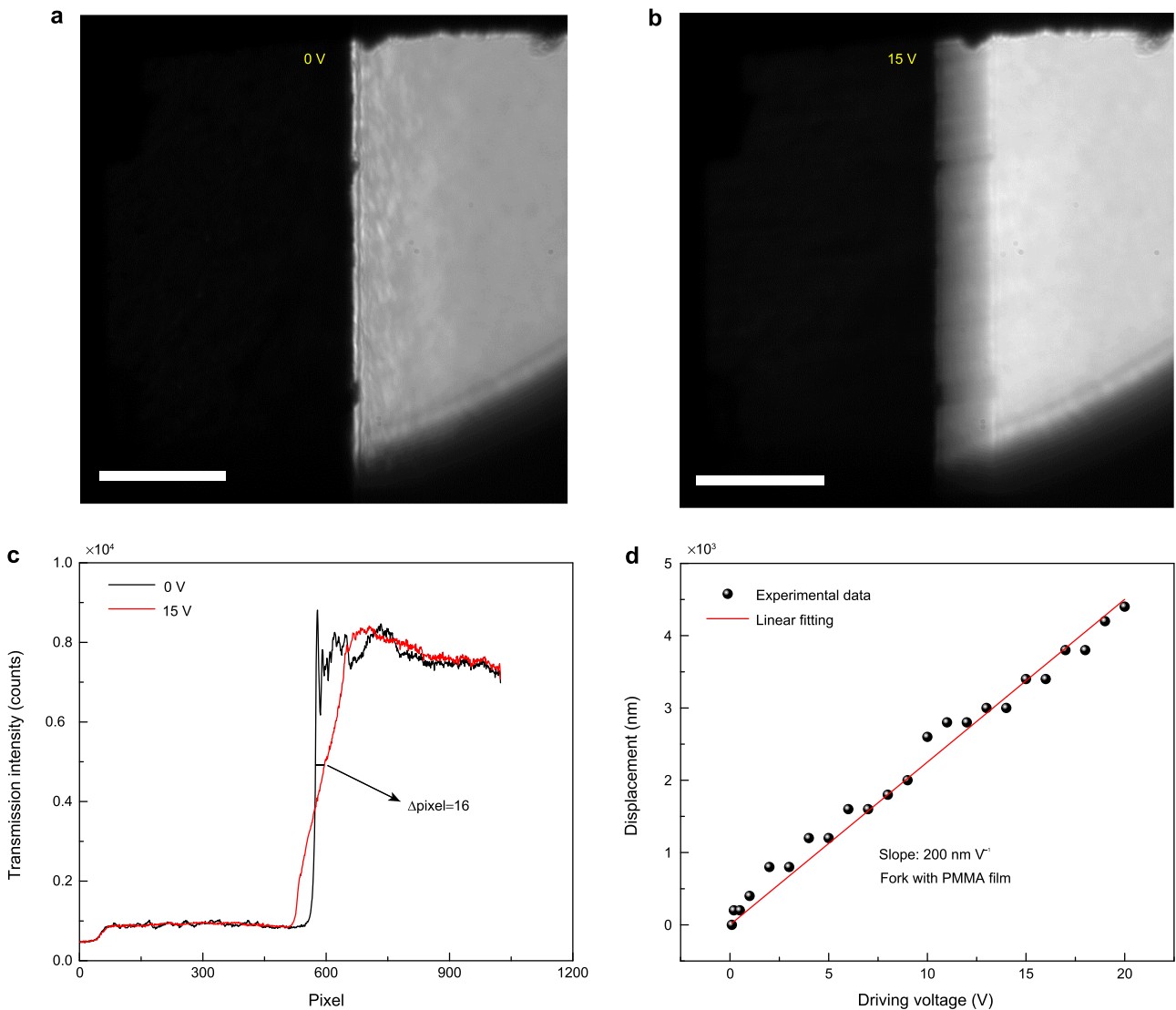

**Fig. 3 Characterization of the displacement of the tuning fork as a function of the driving voltage. a** Bright-field image of the tuning fork prong in the absence of a driving voltage; Scale bar: 50 μm. **b** Bright-field image of the tuning fork prong when driven on resonance with a driving voltage of 15 V. Scale bar: 50 μm. **c** The cross-section perpendicular to the edge of the tuning fork prong. The displacement of the tuning fork is ∼16 pixels at a driving voltage of 15 V. **d** Displacement of the tuning fork as a function of the driving voltage at a resonant frequency of 32.717 kHz. The displacement is linearly proportional to the driving voltage. The red solid line is the linear fitting of the experimental data (solid scattered points) with a slope of ∼200 nm V$^{-1}$.

resonant frequency of the nanoparticle[32]. The nanoparticle confines the electromagnetic field of the incident light into a tiny volume near its surface, leading to an enhancement in light absorption of a molecule, namely, excitation enhancement ($F_{exc}$). Besides, the nanoparticle antenna can also cause a fluorescence enhancement ($F_{em}$) by altering the radiative decay rates ($F_{rad}$) and nonradiative decay rates ($K_{nr}$) of a molecule[28]. The detailed calculations of required parameters are provided in Supplementary Note 10. The calculated $F_{rad}$ and $K_{nr}$ as functions of the molecule-nanorod distance are presented in Supplementary Note 11. $F_{exc}$ and $F_{em}$ as functions of the molecule-tip distance are exhibited in Figs. 4a and b, respectively. $F_{exc}$ decreases and $F_{em}$ increases exponentially as the molecule-tip distance increases. Thus, the total fluorescence enhancement ($F_{tot}$) by the plasmonic nanoparticles originating from the multiplication of $F_{exc}$ and $F_{em}$ and its derivative to the distance ($dF_{tot}/dR$) can be calculated, as illustrated in Figs. 4c and d, respectively. $F_{tot}$ monotonically increases as the distance decreases until a minimum distance of 1.5 nm is reached. Specifically, the largest slope $dF_{tot}/dR$ is

∼700 nm$^{-1}$ when the molecule-tip distance is ∼0.7 nm. Moreover, the fluorescence enhancement factor also strongly depends on the emission wavelength of a single molecule. Then, $F_{rad}$ and $K_{nr}$ as functions of the emission wavelength of the CV molecule were also calculated, as described in Supplementary Note 12. The results are consistent with the extinction spectrum of the gold nanorod, indicating that the calculation method and model are reliable.

The theoretical sensitivity of acoustic detection with a single molecule is mainly limited by the shot noise associated with the fluorescence intensity traces. It can be expressed in the following Eq. (1) (the detailed process for deriving the following equation is provided in Supplementary Note 13):

$$\delta R_{min} = (dR/dF_{tot})/\sqrt{I \cdot t} \qquad (1)$$

The sensitivity is inversely proportional to the first derivative of the total enhancement factor $dF_{tot}/dR$ with a maximum value of ∼700 nm$^{-1}$. Thus, a theoretical acoustic wave detection sensitivity of 10 fm Hz$^{-1/2}$ can be achieved with a single CV molecule at

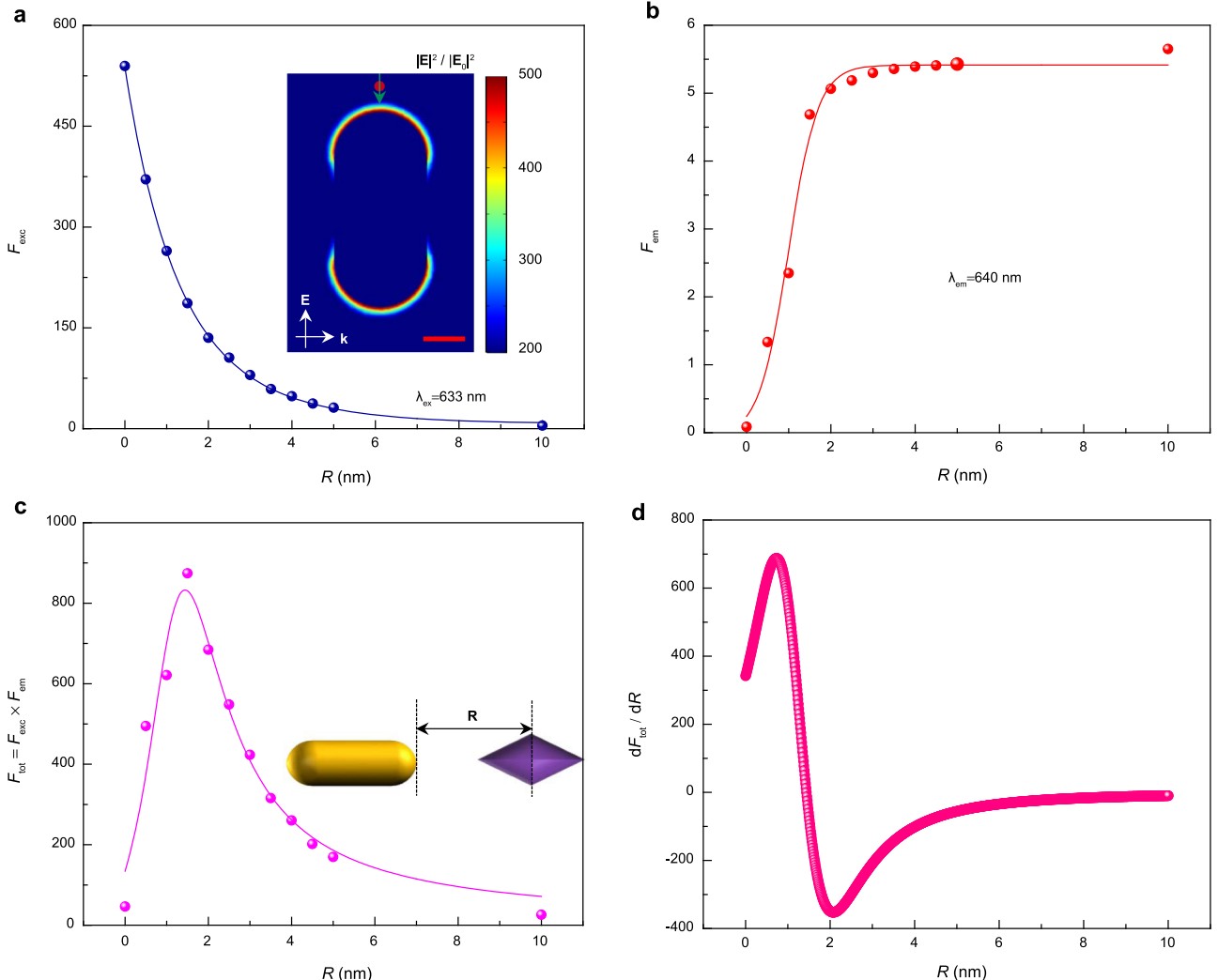

**Fig. 4 Calculation of the fluorescence enhancement factors of a single molecule as functions of the molecule-tip distance. a** Calculation of the excitation enhancement ($F_{exc}$) as a function of the vertical distance ($R$) from the surface of the tip of the gold nanorod. Inset: Map of the calculated near-field electric field around a gold nanorod with a size of 25 × 12 × 12 nm based on the finite-element method (FEM); scale bar: 5 nm. The wave vector direction of the incident light is perpendicular to the longitudinal axis of the gold nanorod, as indicated by k. The localized electric field direction is parallel along the longitudinal axis of the gold nanorod, as indicated by E. The orientation of the molecule was set to be parallel to the longitudinal axis of the gold nanorod, as indicated by the green solid arrow. **b** Calculation of the emission enhancement ($F_{em}$) as a function of the molecule-tip distance ($R$). **c** Calculation of the total fluorescence enhancement ($F_{tot}$) as a function of the molecule-tip distance ($R$). $F_{tot}$ monotonically increases as distance decreases until a minimum distance of 1.5 nm is reached. At shorter distances, however, the fluorescence quenching becomes dominant, and $F_{tot}$ decreases again. **d** The first derivative of $F_{tot}$ as a function of the molecule-tip distance. The largest slope $dF_{tot}/dR$ as a function of the molecule-tip distance is ~700 nm$^{-1}$. The emission wavelength of the point-like dipole was fixed at the emission peak of 640 nm for crystal violet (CV). The solid lines are curves fitted to the scattered points.

a fluorescence intensity of 26,000 counts s$^{-1}$, which is 3 orders of magnitude higher than the experimentally determined value.

**Mechanism of the fluorescence variation**. The fluorescence intensity variation at the resonant frequency of the tuning fork confirms that the tuning fork influences the fluorescence intensity. However, before the conclusion that this signal arises from the acoustic wave, we must exclude all other possible influences, including: (1) the Stark effect. Generally, a single molecule is sensitive to the electric field. The same measurement with a driving voltage at off-resonant frequencies of the tuning fork was conducted to exclude the Stark effect. As described in Supplementary Note 14, the fluorescence intensity variation could be observed only when the driving frequency was resonant with the

tuning fork. This result excludes a direct effect of the electric field, which otherwise would show a constant signal independent of the driving frequency. (2) The displacement of the molecule. The vibration of the tuning fork may result in the displacement of the molecule. There are two reasons for excluding this effect: (I) the displacement of a molecule is orders of magnitude smaller than the diffraction limit, making the fluorescence intensity variation caused by such small displacements negligible. (II) As shown in Supplementary Note 15, two molecules exhibited completely different results in the same area, which would be the same if the fluorescence intensity variation was caused by the displacement of the molecule. (3) The density variation of the film. As given in Supplementary Note 16, the fluorescence intensity variation caused by the density variation was estimated to be ~20 counts, which is about 3 orders of magnitude less than

that caused by the molecule-nanorod distance variation with 13,000 counts s$^{-1}$ under the same condition. Hence, the density variation effect can be excluded. (4) The photoluminescence of the gold nanorod. The photoluminescence of gold nanorods embedded in the PMMA film without CV molecules and their response to acoustic waves were measured, as presented in Supplementary Note 17. The contribution from the photoluminescence of the gold nanorod is also negligible because: (I) the photoluminescence intensity of a gold nanorod is 3–4 orders of magnitude lower than that of a molecule; (II) the photoluminescence has no response to the acoustic wave. (5) Structure deformation of the molecule. The fluorescence variations of a PMMA film doped with a high concentration of CV molecules without gold nanorods were checked during the oscillation of the tuning fork, as indicated in Supplementary Note 18. Any acoustic signals were not observed from the FFT analysis, which directly excludes the deformation of the molecular structure.

It is also possible that the acoustic strain can directly affect the plasmon resonance of the gold nanorod by inducing deformations. Therefore, the response of the scattering intensity of a single gold nanorod to the acoustic strain was investigated, as described in Supplementary Note 19. It was revealed that the scattering of the gold nanorod can also respond to the acoustic wave with a sensitivity as high as 100 pm Hz$^{-1/2}$ which is one order of magnitude lower than that of a single-nanorod system with a sensitivity of 10 pm Hz$^{-1/2}$. The result further confirms that single-molecule fluorescence detection provides the largest sensitivity. Anyway, the molecule-nanorod system can act as nano-detectors of acoustic deformations for both mechanisms.

## Discussion

There are many other emerging techniques for ultrasensitive detection of acoustic displacement, distance, or force, mainly including laser interferometers, AFM nanocantilevers, optical tweezers, MEMS, NEMS, single-electron transistors, and single-molecule electron transfer. For example, high sensitivity detection of displacement can reach fm Hz$^{-1/2}$ based on an interferometer combined with AFM[33]. Even higher sensitivity can be achieved at ultra-low temperatures to be fm Hz$^{-1/2}$, sub-fm Hz$^{-1/2}$, and am Hz$^{-1/2}$ by SQUIDs, SQL, and SAW, respectively[5,34,35]. Nevertheless, all of the techniques, though with high sensitivity, are limited to low temperatures. At room temperature, Ohlinger et al.[9] reported a displacement sensitivity of 3 nm Hz$^{-1/2}$ with a single gold nanoparticle trapped by optical tweezers, which can be improved to fm Hz$^{-1/2}$ level by a dual-beam optical tweezer in air[8]. Yang et al.[36] demonstrated a distance sensitivity of 30 pmHz$^{-1/2}$ for structural variation of flavin-tyrosine in a single protein molecule probed by single-molecule electron transfer. These techniques can be applied at room temperature but the complicated setups and the environment requirements limit their applications. Moreover, the detection bandwidth of these techniques is relatively narrow.

In our work, the molecule-nanorod acoustic detector with a size of 40 nm can detect longitudinal, compression, and transverse shear waves modulating the distance between the molecule and gold nanorod. The experimental displacement sensitivity of 10 pm Hz$^{-1/2}$ could be possibly improved to several orders of magnitude by optimizing the experimental conditions. The detection bandwidth can reach dozens of GHz, with potential application in the detection of high-frequency vibrations with both high spatial resolution and detection sensitivity. Besides, the low laser power for exciting a single molecule can effectively prevent the undesired heating or other disturbance by absorption of light in the readout process of quantum states owing to the light focus and the plasmonic enhancement. Furthermore, the significant advantage of this technique is that all the electronic

noise of analogous detectors and amplifiers is filtered out by photon counting. FFT of the fluorescence intensity trace works as a virtual lock-in amplifier for the shot-noise-limited signals from the photon counter.

These experimental and theoretical results demonstrate a large room to improve the detection sensitivity by optimizing the system. The most crucial factors determining the detection sensitivity are the molecule-nanorod distance and the orientation of the system. In this work, both the distance and orientation of the molecule and gold nanorod in the film were random, limiting the detection sensitivity. According to the estimation, most of the detected molecules were located 2–7 nm away from the nanorod, still "far" from the optimized distance of 0.7 nm. Additionally, the best sensitivity can be obtained only when the orientation is optimized at which the acoustic displacement induces the largest variation in molecule-nanorod distance. On the contrary, when the acoustic strain is perpendicular to the molecule-nanorod system, we expect a less efficient detection of the acoustic signal. Other experimental conditions could also affect the detection sensitivity, including environmental noise, the collection efficiency of the fluorescence photons, the detection efficiency of the photodiode, and the possible blinking behavior of the molecule. The proposed molecule/nanorod nano-probe system can realize sensitive detection for acoustic waves at room temperature.

In the future, the detection sensitivity can be further improved by either optimizing the system or enlarging the plasmonic effect. For example, a chemical linker such as DNA can be employed to accurately control the distance between the molecular probe and the gold nanorod. A "hot spot" can be formed by two gold nanorods, providing a much higher localized electric field, in which the molecule could be a nano-acoustic detector with much higher sensitivity.

In summary, the demonstrated optical detection mechanism for acoustic waves is free of electromagnetic interference and presents potential applications in various areas. The proposed nano-acoustic detector with a small size of 40 nm achieves high sensitivity and spatial resolution, and can be placed near the acoustic sources, nano-electromechanical systems, or quantum mechanics to read out their vibrational frequency, amplitude, and quantum states. The direction sensitivity of such a nano-acoustic detector provides potential applications for accurately localizing and tracking acoustic sources. Inspired by super-resolution microscopy, such a nano-acoustic detector could also offer fundamental elements for high-resolution acoustic microscopy by localizing the molecules at the nanometer scale.

## Methods

**Sample preparation.** Gold nanorods with average dimensions of (35±5) × (16±1) nm were purchased from Beijing Zhongkeleiming Daojin Technology Co., Ltd, and dissolved in DMSO. Crystal violet (CV), poly (methyl methacrylate) (PMMA, average molecular weight: M$_n$≈996,000) and Dimethyl sulfoxide (DMSO, AR) were purchased from Sigma-Aldrich and Macklin, respectively, and used without further purification.

First, we prepared the stock solution of CV with concentration of $1.0 \times 10^{-5}$ mol L$^{-1}$ (M) and 40 mg mL$^{-1}$ PMMA matrix polymer both dissolved in DMSO. Next, we diluted the stock solution of CV 10-fold for 2 times gradually using 40 mg mL$^{-1}$ PMMA to achieve approximate single-molecule concentration. Also, we diluted the stock solution of gold nanorods with concentration of $2.0 \times 10^{-10}$ M (0.1 mg mL$^{-1}$) for 5 times using 40 mg mL$^{-1}$ PMMA. Finally, we mixed the diluted CV solution and gold nanorods solution with volume ratio of 1:1. We dropped 30 μL of as-prepared mixed solution on the glass coverslip and spin-coated at 820 rpm. The spin-coated film was then annealed at 70 °C for 25 min in order to remove the solvent thoroughly. After that, we stripped the PMMA film from the glass coverslip using the needle of injection syringe to scratch the thin film and then stuck it to both prongs of the tuning fork with glue. Preparation of all samples took place under ambient conditions.

**Material characterizations.** The absorption spectra of the CV in DMSO solution and in the PMMA film as well as of the gold nanorod suspension were measured on a UV-vis spectrophotometer (Shimadzu UV-3600). Morphologies of gold nanorods dropped and dried on a silicon chip were analysed using scanning

electron microscopy (SEM, FEI NanoSEM-450) operating at an accelerating voltage of 15.0 kV.

**Measurement of the absolute fluorescence quantum yield.** The quantum yield (QY) was measured using the Edinburgh FLS 1000 spectro-fluorimeter. The absolute fluorescence QY is the ratio of the number of photons emitted to the number of photons absorbed:

$$QY = \frac{N^{em}}{N^{abs}}$$

For liquid samples, we used DMSO solvent as the control sample to adjust the number of photons to 1,000,000 at 600 nm, and then adjusted the photons of sample to 800,000 by changing the concentration of CV in DMSO for the QY measurement. For film samples, we used $BaSO_4$ as the control sample to adjust the number of photons to 1,000,000 at 600 nm, and then replaced with CV in PMMA film for QY measurements.

**Optical measurements.** Our home-built wide-field fluorescence microscope for single-molecule imaging and spectroscopy is shown in Supplementary Note 1. In details, a semiconductor-pumped solid-state laser of 633 nm (MRL-III-633-200mW, Changchun New Industries Optoelectronics Tech Co., Ltd., China) after passing a 633-nm Laser-line filter (LL01-633, Semrock) was used as the excitation light source for acoustic detection measurements. A fluorescence microscope (Olympus IX73) with a dry objective (×40, NA=0.6, Olympus LUCPlanFl) focused the linearly polarized light to a circular spot with excitation area of $7.5 \times 10^{-5}$ cm$^2$. The excitation power density at 633 nm was calculated to be 50 W cm$^{-2}$. Single-molecule fluorescence after passing through a 635 nm dichroic mirror (FF635-Di01, Semrock) and a 655 nm long-pass filter (ET655lp, Chroma) in that order was split by a beam splitter mirror (Daheng optics, China), then collected and imaged on a single-photon counting system (TCSPC, Picoharp 300) and EMCCD camera (iXon Ultra 888, Andor), respectively. A 532 nm CW diode laser provided an excitation intensity of 10 W cm$^{-2}$ to test the photostability of CV molecules at the ensemble and single-molecule levels. A transmission grating (Newport, 150 lines mm$^{-1}$) was put in front of the camera for fluorescence spectrum measurements. Fluorescence lifetime measurements were carried out using a time-correlated single-photon counting system (TCSPC, Picoharp 300) with a super continuous laser (Fianium SC-400) as the excitation light source. In our experiment, we used a signal generator (DG4202, RIGOL) to modulate the driving frequency of the tuning fork carrying the PMMA film and observed the amplitude change of the sine wave response with an oscilloscope (MDO3034, Tektronix) to find out the resonant frequency of the tuning fork. As a result, the tuning fork can be stretched and compressed periodically at the resonant frequency, leading to a periodic change in the distance between a single molecule and a gold nanorod.

## Data availability

The data that support the findings of this study in Main Text are available from the Source Data file. The data support the findings of this study in Supplementary Information file are available from the corresponding authors upon request due to the large size of data files. Source data are provided with this paper.

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

## Acknowledgements

This work was supported by the National Natural Science Foundation of China (NSFC no. 22073046). This work was also supported by State Key Laboratory of Analytical Chemistry for Life Science (SKLACL2217) and the Fundamental Research Funds for the Central Universities (020514380256). X.L. acknowledges a Ph.D. grant from China Scholarship Council. We thank Dr. Panke Zhang for his kind help in processing the raw

tuning fork. We also thank Prof. Ye Tian and Dr. Shuang Wang in the College of Engineering and Applied Sciences for their kind help with preparation of the DNA linker between a gold nanosphere and a gold nanorod.

## Author contributions

M.X. carried out the experiments, performed the theoretical simulation and wrote the manuscript. H.L. performed the distance control between a gold nanosphere and a gold nanorod by DNA linker. S.W., D.H. and Z.W. helped with the setup. X.L. helped with the theoretical simulation. Y.D. helped the displacement characterizations of the tuning fork. W.Y. did the resonant spectrum of the tuning fork. S.F. and W.X. prepared the gold bowties structure. C.-L.T., D.X. and X.-J.W. did the TEM characterizations. B.W. and S.L. did the fluorescence quantum yield measurements. M.O. and Y.T. conceived the idea and supervised the project. All authors were involved in writing and revising process of the manuscript.

## Competing interests

The authors declare no competing interests.
