## [Peer Review File · Nature Communications]

Ultrasensitive Detection of Local Acoustic Vibrations at Room Temperature by Plasmon-Enhanced Single-Molecule FluorescenceEditorial Note: This manuscript has been previously reviewed at another journal that is not operating a transparent peer review scheme. This document only contains reviewer comments and rebuttal letters for versions considered at Nature Communications.

REVIEWER COMMENTS

Reviewer #4 (Remarks to the Author):

Most of the remarks and questions from my previous report at another journal were correctly addressed by the authors. I would like nevertheless to clarify a point regarding the sensitivity to vibrations.

1- To me the experimental detection sensitivity is given by the background noise level of the FFT spectrum (which I assume is dominated by the fluorescence shot noise). This gives a signal to noise ratio of 1. To give a noise sensitivity per root hertz, this value should be corrected to give the noise level in a one second measurement time. Is this the procedure used by the authors ?

2- I still didn't get how the FFT of the fluorescence rate is calibrated in meters. The proper calibration would involve the measurement of the fluorescence variation for a known static displacement. This would yield a calibration static factor in counts/metres which could be extrapolated to the resonant frequency if the photon detector has a flat frequency response. Few words on the calibration procedure should be given in the manuscript since the extracted sensitivity depends on it.

3- Regarding the measurement bandwidth, if the Nyquist's criterion is naively applied it is limited by the fluorescence rate. It yields a bandwidth of ~ 4.3 kHz in your case (6 photon in average per oscillation). There is two ways to circumvent this limitation:

- A repeated pump probe measurement with a monochromatic signal. Since the fluorescence photon emission is random, the trajectory can be reconstructed over many realisation. This technique won't work for noise measurement (mechanical Brownian motion).
- Measuring correlations between the photon arrival time (the auto-correlation function). This is the way a spectrum analyser works. The only limitation is quality factor of the mechanical resonance.

Given these advices are adressed, I recommend publication of this article in Nature Communications.

We thank the reviewer for his/her valuable and constructive criticisms. We have revised our manuscript according to the comments. Please find our point-by-point responses as follows.

Reviewer #4 (Remarks to the Author):

Most of the remarks and questions from my previous report for another journal were correctly addressed by the authors. I would like nevertheless to clarify a point regarding the sensitivity to vibrations.

1- To me the experimental detection sensitivity is given by the background noise level of the FFT spectrum (which I assume is dominated by the fluorescence shot noise). This gives a signal to noise ratio of 1. To give a noise sensitivity per root hertz, this value should be corrected to give the noise level in a one second measurement time. Is this the procedure used by the authors ?

Response: Yes, this is exactly what we did for the sensitivity calculation. The noise sensitivity with a unit of $\text{pm}/\text{Hz}^{1/2}$ has been normalized to 1 s. Based on the raw data, the detectable displacement was calculated to be 0.7 pm. Considering the integration time of 200s, we get the sensitivity of $10 \text{ pm}/\text{Hz}^{1/2}$.

2- I still didn't get how the FFT of the fluorescence rate is calibrated in meters. The proper calibration would involve the measurement of the fluorescence variation for a known static displacement. This would yield a calibration static factor in counts/metres which could be extrapolated to the resonant frequency if the photon detector has a flat frequency response. Few words on the calibration procedure should be given in the manuscript since the extracted sensitivity depends on it.

Response: We thank the review for pointing out this issue. The FFT signal was calibrated to meter based on the following procedure: 1) We measured the displacement of the tuning fork arm at different driving voltages and got the

calibration factor between displacement and voltage to be 13 pm/V. 2) We measured the fluorescence intensity trace at different driving voltages and analyzed the FFT. The smallest detectable FFT signal was obtained with the driving voltage of 0.05 V corresponding to 0.7 pm. Then we calibrated other FFT signals according to this value.

However, after careful consideration, we think such calibration is necessary only when we need to measure an unknown displacement with our detector. It does not provide extra information for this work. Thus we decided to remove this calibration and directly use the FFT signal with arbitrary unit to show the smallest detectable displacement. We have updated the Fig. 2 and revised the description in the Supplementary Information.

3- Regarding the measurement bandwidth, if the Nyquist's criterion is naively applied it is limited by the fluorescence rate. It yields a bandwidth of ~ 4.3 kHz in your case (6 photon in average per oscillation). There is two ways to circumvent this limitation:

- A repeated pump probe measurement with a monochromatic signal. Since the fluorescence photon emission is random, the trajectory can be reconstructed over many realisation. This technique won't work for noise measurement (mechanical Brownian motion).
- Measuring correlations between the photon arrival time (the auto-correlation function). This is the way a spectrum analyser works. The only limitation is quality factor of the mechanical resonance.

Response: We agree with the reviewer that at least 6 data points are needed per oscillation which determines the bandwidth. However, recording 6 data points does not require 6 photons per oscillation in our case because we measure many cycles of the oscillation in one second. This method used exactly the same mechanism as the pump probe measurements as suggested by the reviewer. To achieve such broad bandwidth, we need to excite the molecule many times to collect enough photons. Actually, we have shown the ability to measure vibration signal with frequency of 32 kHz provided by the tuning fork as discussed in the manuscript. After careful

consideration, we believe the bandwidth depends on the time resolution of our photon-counting instrument, which is 4 ps, corresponding to a bandwidth of 40 GHz when considering 6 points per oscillations according to Nyquist's criterion. We also admit that this bandwidth is not available for the real-time measurements which was determined by the fluorescence rate. We added one sentence to clarify the real-time bandwidth on page 9 lines 199-202.

Given these advices are addressed, I recommend publication of this article in Nature Communications.

REVIEWERS' COMMENTS

Reviewer #4 (Remarks to the Author):

I am convinced by the detailed response of the authors to my previous questions as well as the changes they made in the manuscript. I recommend the article "Ultrasensitive Detection of Local Acoustic Vibrations at Room Temperature by Plasmon-Enhanced Single-Molecule Fluorescence" by Dr. Tian for publication in Nature Communications.

Reviewer #4 (Remarks to the Author):

I am convinced by the detailed response of the authors to my previous questions as well as the changes they made in the manuscript. I recommend the article "Ultrasensitive Detection of Local Acoustic Vibrations at Room Temperature by Plasmon-Enhanced Single-Molecule Fluorescence" by Dr. Tian for publication in Nature Communications.

Response: We thank the reviewer for his/her positive recommendation and great contribution for the improvement of our manuscript.